https://doi.org/10.1038/s42004-020-00373-2　**OPEN**

# Silylium ion mediated 2+2 cycloaddition leads to 4+2 Diels-Alder reaction products

Heng-Ding Wang [1,2] & Hong-Jun Fan [1,2✉]

The mechanism of silver(I) and copper(I) catalyzed cycloaddition between 1,2-diazines and siloxy alkynes remains controversial. Here we explore the mechanism of this reaction with density functional theory. Our calculations show that the reaction takes place through a metal ($Ag^+$, $Cu^+$) catalyzed [2+2] cycloaddition pathway and the migration of a silylium ion [triisopropylsilyl ion ($TIPS^+$)] further controls the reconstruction of four-member ring to give the final product. The lower barrier of this silylium ion mediated [2+2] cycloaddition mechanism (SMC) indicates that well-controlled [2+2] cycloaddition can obtain some poorly-accessible IEDDA (inverse-electron demand Diels-Alder reaction) products. Strong interaction of $d^{10}$ metals ($Ag^+$, $Cu^+$) and alkenes activates the high acidity silylium ion ($TIPS^+$) in situ. This π-acid ($Ag^+$, $Cu^+$) and hard acid ($TIPS^+$) exchange scheme will be instructive in silylium ion chemistry. Our calculations not only provide a scheme to design IEDDA catalysts but also imply a concise way to synthesise 1,2-dinitrogen substituted cyclooctatetraenes (1,2-NCOTs).

[1] State Key Laboratory of Molecular Reaction Dynamics, Dalian National Laboratory for Clean Energy, Dalian Institute of Chemical Physics, Chinese Academy of Sciences, Dalian 116023, China. [2] University of Chinese Academy of Sciences, Beijing, China. ✉email: fanhj@dicp.ac.cn

Catalyzed [4+2] inverse-electron demand Diels–Alder reactions (IEDDA) remain challenging owing to biased activation of more electron-rich substrates[1–3]. Most catalytic approaches center on modulating two π-electron substrates for a better LUMO-HOMO matching according to FMO[4]. Lewis acid-catalyzed IEDDA between electro-rich alkynes (such as siloxy alkynes, ynamines) and aromatic diene (such as phthalazines) often requires relatively harsh conditions[5,6].

[2+2] cycloaddition have been earned broad interest in recent years[7,8]. Four-member ring [2+2] cycloaddition product between alkynes and aromatic ring can undergo ring expansion to form cyclooctatetraene similarities (COTs), COTs can further undergo ring contraction and retro [2+2] to cover [4+2] Diels–Alder product[9–11]. Higashino and coworkers reported that substituted 1,2-diazines with ynamines were shown to proceed both [2+2] cycloaddition and [4+2] IEDDA cycloaddition, the [2+2] adduct followed with ring expansion to give diazacyclooctatetraene derivatives. They obtained [4+2] IEDDA product and diazacyclooctatetraene derivatives with comparable yield[12,13] However, these schemes were limited by high reaction barrier and low chemo-selectivity[14]. To best our knowledge, there remains no report of [2+2] cycloaddition pathway which can exceed [4+2] Diels–Alder pathway to produce IEDDA products so far.

In 2012, Rawal and coworkers reported that silver (I) can efficiently catalyze phthalazines (**sub1**) and siloxy alkynes (**sub2**) to get 3-substituted 2-naphthol silyl ethers (**3**) in the room temperature when paired with 2,2-bipyridine as ligand. In 2014 they further discovered that copper(I) can supplant silver (I) to catalyze this reaction (Fig. 1a)[15,16]. For the strong Lewis acid (such as TiCl$_4$, SnCl$_4$, Sc(OTf)$_3$ etc.) are ineffective of this reaction, they proposed that the high affinities of copper(I) and silver(I) to alkynes may take key roles in this reaction. They also found an interesting phenomenon that relatively electron richer ethoxy alkynes **2c** and ynamide **2d** (Fig. 1b) cannot react with phthalazines (**sub1**) in the same condition. The author proposed two pathways, the first pathway is coordination silver ion of phthalazines (**sub1**) rends it more electro deficiency (concerted [4+2] mechanism), and another plausible pathway which proceed through silver(I) induced nucleophilic attack of siloxy alkynes (**sub2**) to phthalazines (**sub1**) (stepwise [4+2] mechanism).

Recently Avcı et al. have explored silver(I) catalyzed [4+2] mechanism of this reaction[17]. Using a reduced model, they proposed the reaction can undergo through [4+2] stepwise mechanism with respect to ligand coordinated silver(I) (**L-Ag$^+$**), **sub1** and **sub2** as reference point. With the same model and method, we found the barrier of [4+2] reaction may be underestimated in their work and [4+2] pathway is not favored in the room temperature(see details in Supplementary Note 1).

Note that there were quite many examples of d$^{10}$ metal-induced nucleophilic reaction[18–20]. We proposed silylium ion [triisopropylsilyl ion (TIPS$^+$)] mediated [2+2] cycloaddition (SMC) mechanism which accords with all experimental observations (Fig. 2a).

In our proposed silylium ion mediated [2+2] cycloaddition mechanism (SMC), break aromaticity of **sub1** to form four-member ring **int3** is essential step, which is a d$^{10}$ metal [Ag(I) or Cu(I)] catalyzed [2+2] cycloaddition reaction. TIPS$^+$ transfer to the tertiary amine of **int3** further induce heterolysis of C-N bond which lead to metal coordinated ketene species **int4a-1**. Ketene species **int4a-1** can easily undergo ring closure to form **int5a** which is very facile to release N2, TIPS$^+$ and metal(I) phenate complex **int7**. TIPS$^+$ combined with **int7** gives final product.

As Rawal pointed, high affinity of d$^{10}$ metals (Ag$^+$ or Cu$^+$) to alkynes and alkenes take crucial role in this reaction, which triggers the [2+2] cycloaddition reaction and migration of silylium ion (TIPS$^+$) consecutively. If the silylium ion transfer step is inhibited and only [2+2] cycloaddition take place, this reaction can lead to four-member *11b H-Cyclobuta [2,1,α]phthalazine (CBP)* which further undergo ring expansion to form 1,2-dinitrogen substituted eight-member ring 1,2-NCOT (Fig. 2b). 1,2-NCOT is interesting similarities of cyclooctatetraene (COT) and its reactivity remains almost unknown. This reaction may imply a very concise method to synthesis 1,2-NCOT which is kinetic stable according to our calculation. Our mechanism also provides an alternative solution for other alkynes which are inert to IEDDA, such as ynamines and ynamides etc[5]. If the protection group of ynamide **2d** (Fig. 1b) changed into silicane group, this SMC pathway may extend to nitrogen-substituted alkynes. Silylium ion advantaged in its prominent high electrophilicity which showed excellent catalyze performance compared with traditional Lewis acid[21–27]. In this work, π-acidic d$^{10}$ metals activate silylium ion in situ which then act as catalyst, this π-acid to hard acid scheme may also enlighten in silylium ion chemistry for most of silylium ion catalyzed reactions requires high acidity silylium ion prepared before reaction or generated in situ which need extremely active carbocation reagent[22,23,28].

## Results

**Formation of Π-complexes M-int1(M = Ag or Cu).** In the silver case, coordination of 2,2′-bipyridine (bpy) to the catalyst salt AgOTf gives the bi-coordinated active center **L-Ag$^+$** is relatively exothermic by −23.1 kcal/mol (Fig. 3a). The next step is

a. Silver(I) and Copper(I) Catalyzed iEDDA reported by Rawal et al

b. Unreactive alkynes

**Fig. 1 An overview of silver(I) and copper(I) catalyzed IEDDA. a** Reaction conditions; **b** Unreactive alkynes.

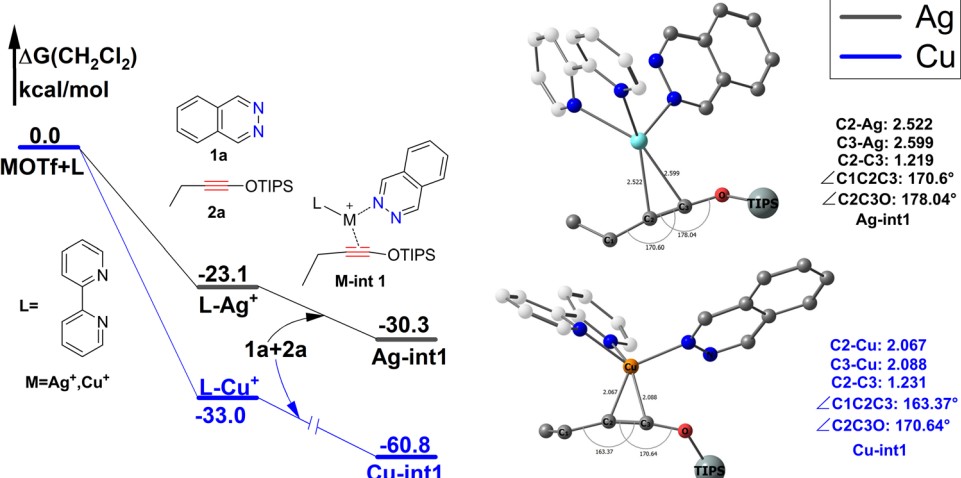

a. Our Proposed Silylium ion Mediated 2+2 Cycloaddition Mechanism

**Fig. 2 An overview of our proposed silylium ion mediated [2+2] cycloaddition mechanism. a** Formation of IEDDA product; **b** Formation of 1,2-NCOT.

**Fig. 3 Reaction profiles of formation π-complexes Ag-int1 and Cu-int1.** Gibbs free energies are in kcal/mol. The hydrogen atom was omitted and triisopropylsilyl (TIPS) group was simplified to big ball for clarity. **a** Reaction profile; **b** Optimized geometries.

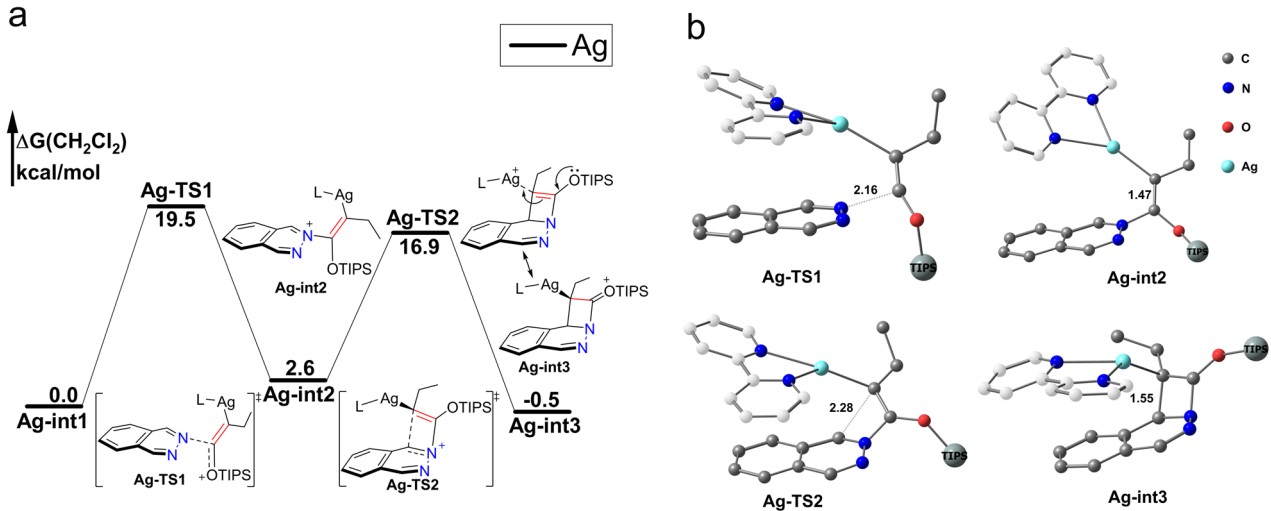

**Fig. 4 Formation silver coordinated four-member ring complex Ag-int3.** L refers to 2,2′-bipyridine (bpy). Gibbs free energies are in kcal/mol. The hydrogen atom was omitted, triisopropylsilyl (TIPS) group was simplified to big ball and the carbon atom of ligand [2,2′-bipyridine (bpy)] was white cloaked for clarity. **a** Reaction profile; **b** Optimized geometries.

coordination of phthalazine **1a** and siloxy alkynes **2a** with bared active site **L-Ag$^+$** to obtain π-complex **Ag-int1** which is exothermic by −30.3 kcal/mol.

In the copper(I) catalyzed IEDDA reaction, formation of π-complex **Cu-int1** is very exergonic (−60.8 kcal/mol). The higher exothermic energy of formation **Cu-int1** may be caused by relatively small diameter and high positive charge density of copper(I). **Cu-int1** is more structurally compact compared with **Ag-int1**, and siloxy alkyne **2a** coordinate more stronger with copper(I) atom (Fig. 3b). In both **Cu-int1** and **Ag-int1**, the siloxy **2a** significant deviations from linearity and in **Cu-int1** triple bond C2-C3 elongate from 1.21 Å to 1.23 Å while in **Ag-int1** C2-C3 bond elongate to 1.22 Å. The bond angle of ∠C1C2C3 twist from 180° to 163.37° in **Cu-int1** and 170.6° in **Ag-int1**. Current literature have revealed quite a few well characterized copper(I) and silver(I) coordinated alkynes π-complexes and confirm that copper(I) coordinate stronger with alkynes than silver(I) in these π-complexes[29]. The H1 NMR and C13 NMR experiment also confirm evident interactions between siloxy alkynes with Ag(I) in DCM[30].

Note that π-complexes **Ag-in1** and **Cu-int1** are the most stable intermediate before the reaction, we choose **Ag-int1** and **Cu-int1** as zero point for silver(I) and copper(I) catalyzed IEDDA respectively in the following discussion.

**Silver catalyzed [2+2] Cycloaddition.** Thermal constrains of orbital symmetry pose much challenge to direct [2+2] cycloaddition. Further, direct [2+2] cycloaddition of **1a** and **2a** needs to broke the aromaticity of the phthalazine ring. Our calculation shows that the barrier of this process as high as 54.6 kcal/mol. Transition metal offers the prospect of promoting the [2+2] cycloaddition through active alkynes by virtue of valence d-orbitals and lower energy of metallacyclic intermediates. Metal-catalyzed [2+2] reaction was also investigated by theory[31,32].

Similar with most of metal-catalyzed cycloaddition reaction, nucleophilic addition of phthalazines **1a** with siloxy alkene **2a** activated by **L-Ag$^+$** is the initial step (Fig. 4a).

Our calculation shows that formation of ethylene silver species **Ag-int2** is energetically accessible in the room temperature with a barrier (**Ag-TS1**) 19.5 kcal/mol (Fig. 4a), lower barrier of this process is mainly because the lone pair of oxygen atom in the **Ag-TS1** conjugate with π orbitals of alkyne to stabilize the positive

charge of silver(I). The NBO charge distribution shows the positive charge transferred from silver(I) to the nitrogen atom from **Ag-int1** to **Ag-int2** (N atom from −0.36 to −0.12, Ag atom 0.71 to 0.52). The phthalazine **1a** remains nearly planar in the ethylene silver species **Ag-int2**, which indicate that the aromaticity of **1a** is not completely destroyed in this step (Fig. 4b). An alternative step to obtain ethylene silver species occur via the attack of the siloxy alkyne from C2 point which is energetically unfavored with a barrier 32.4 kcal/mol.

The **Ag-int2** transform into silver(I) coordinated four-member ring complex **Ag-int3** is energetically accessible with a barrier (**Ag-TS2**) 14.3 kcal/mol. Compared with direct [2+2] cycloaddition between **1a** and **2a**, silver(I) catalyst can efficiently lower the barrier of this process (54.6 kcal/mol vs 19.5 kcal/mol). Experiment also confirmed that silver(I) and copper(I) can catalyze [2+2] cycloaddition of siloxy alkynes and substituted alkenes with quite good yield in room temperature[16,30].

**Silylium ion mediated four-member ring rearrangement (path a).** Direct migration of triisopropylsilyl ion (TIPS$^+$) to form quaternary ammonium species **Ag-int4a** is energetically unfavored with a barrier (**Ag-TS3a**, see Supplementary Data 20) 31.9 kcal/mol. Silylium ion is notoriously unstable, and TIPS$^+$ ion dissociate directly through SN1 pathway is also energetically unfavored with a barrier as high as 49.1 kcal/mol. However, against our chemical intuition, dissociate of TIPS$^+$ through OTf$^-$ anion SN2 nucleophilic attack is quite easy with a barrier (**Ag-TS3**) only 14.9 kcal/mol (Fig. 5a). The high affinity of silver(I) with alkene anion compensate majority formation energy of TIPSOTf and made this process accessible with only 5.8 kcal/mol. The stabilized energy of **L-Ag$^+$** in the **Ag-int4b** as high as 33.1 kcal/mol. The strong interaction of silver(I) with alkene made **Ag-int4b** a very good leaving group. Commonly, weak Lewis base such as SO$_4{}^-$, OTf$^-$ are not good nucleophilic reagent, this situation changed when confront with good leaving group. Further, for the relative larger size of silicon atom compared with carbon atom, the steric protection effect of bulky substituents is less effective and can allow five coordinated structure, which means that many dissociative SN1-type reactions in organic chemistry can proceed through associative-SN2 mechanism with pentacoordinate transition state in silicon chemistry[33].

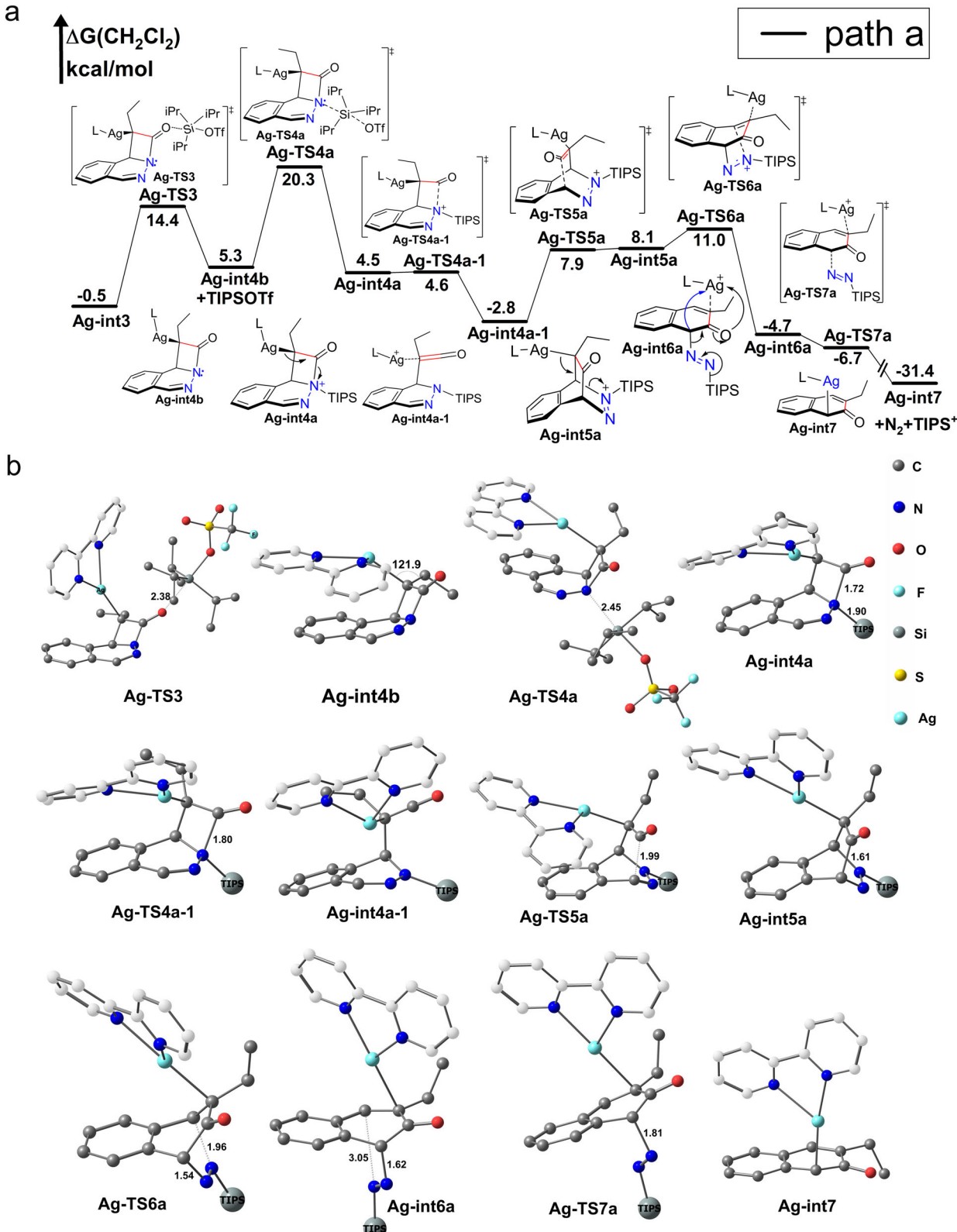

**Fig. 5 Silylium ion mediated four-member ring rearrangement to give Ag-int7 (*path a*).** L refers to 2,2'-bipyridine(bpy). Gibbs free energies are in kcal/mol. Hydrogen atom was omitted and carbon atom of ligand [2,2'-bipyridine(bpy)] was white cloaked for clarity. Triisopropylsilyl (TIPS) is simplified into big ball except desilication process and versa. **a** Reaction profile; **b** Optimized geometries.

TIPSOTf is regarded as anion stabilized silylium ion from a synthetic perspective, which is highly active reagent for amino and hydroxy protection. TIPS$^+$ can easily transferred to nitrogen atom of tertiary amine in **Ag-int4b** to form quaternary ammonium species **Ag-int4a** through SN2 nucleophilic addition with a barrier (**Ag-TS4a**) 15.0 kcal/mol. In the quaternary ammonium species **Ag-int4a**, C-N bond is very easy to relax high tension of four-member ring and form silver(I) coordinated ketene species **Ag-int4a-1** with a barrier (**Ag-TS4a-1**) only 0.1 kcal/mol. In this process positive charge transferred from nitrogen and silicon atoms (N-Si) to silver and carbon atoms (Ag-C). (NBO charge distribution shows that charge of N-Si change from 1.58 to 1.47, while Ag-C atoms change from 0.19 to 0.32). **Ag-int4a-1** can easily undergo ring closure to form six-member **Ag-int5a** with a barrier (**Ag-TS5a**) 10.7 kcal/mol, the positive charge returns back from Ag-C to N-Si. (NBO charge distribution shows that charge of N-Si change from 1.47 to 1.79, and Ag-C change from 0.32 to 0.16). From four-member ring intermediate **Ag-int4a** to six-member ring intermediate **Ag-int5a**, there actually experience twice electron extraction competition between Ag-C and Si-N which is dominated by TIPS$^+$ and **L-Ag$^+$**. In the **Ag-int4a** the high positive charge of Si-N extracts the electron from electron-rich Ag-C bond, which lead to ring open of **Ag-int4a** to form **Ag-int4a-1**. The driving force of this step is high positive charge of Si-N and high tension of four-membered ring of **Ag-int4a**. The barrier from silver(I) coordinated ketene species **Ag-int4a-1** to six-member ring **Ag-int5a** is 10.7 kcal/mol. Although high reactivity of ketene species and formation of N=N bond contribute to this step, this process is endothermic due to high Lewis acidity character of silylium ion, the driving force of this step is extreme high exothermic of release N$_2$. Generally, release N$_2$ is highly exothermic and go through concerted reaction pathway[34,35]. In this reaction, release N$_2$ go through stepwise mechanism which can be attributed to high positive charge of silylium ion (TIPS$^+$). In the **Ag-int5a** the length of C-N bond directly links to silylium ion (TIPS$^+$) is 1.53 Å, another C-N bond is 1.45 Å, which is mainly caused by high positive charge of TIPS$^+$. The relatively longer C-N bond induced by TIPS$^+$ in the **Ag-int5a** indicate its relatively easy heterolysis character. Further, the electron-rich C-Ag bond can donate its electron to the C atom after heterolysis, this is very important driving force of C-N bond heterolysis. Actually, this heterolysis process is also an electron extraction competition between Ag-C and Si-N, the positive charge transferred from N-Si to C-Ag this time (NBO charge distribution shows that electron changed from 0.16 to 0.59 in C-Ag and 1.79 to 1.58 in N-Si from **Ag-int5a** to **Ag-int6a**). Because of these reasons, the barrier for heterolysis C-N bond and stepwise N2 release is only 2.9 kcal/mol. If there are no TIPS$^+$ in this structure, the reaction can go through concerted mechanism with a barrier (see **Ag-TS7b** in Fig. 6b) 6.7 kcal/mol. The silylium ion stabilized dinitrogen intermediate **Ag-int6a** is very facile to release N$_2$, TIPS$^+$ and **Ag-int7** (the negative free energy barrier of this step come from entropy and single point energy correction). TIPS$^+$ combination with **Ag-int7** to give final product is highly exothermic (−60.3 kcal/mol).

**Ketone-enol tautomerism path way (path b)**. In the **Ag-int4b**, the α-carbon is close to sp3 hybridization. The C-Ag-C angle is 121.9° (Fig. 5b), Like its α-H substituted ketone analogs, the intermediate **Ag-int4b** can go through ketone-enol tautomerism style intramolecular isomerization (Fig. 6) to obtain **Ag-int5b-1** which is also the rate determine step of *path b* (23.9 kcal/mol). Direct dissociation of L-Ag$^+$ from **Ag-int4b** is energetically unfavored with a barrier 33.1 kcal/mol (see Supplementary Fig. 4.).

**Copper(I) catalyzed IEDDA with SMC mechanism**. In the copper(I) catalyzed IEDDA, the rate determine barrier (**Cu-TS1**)

of formation of four-member **Cu-int3** is 26.8 kcal/mol (Fig. 7a), which is higher than silver(I) case (19.5 kcal/mol). After formation of **Cu-int4b**, only *path a* is energetically favored with a barrier 7.6 kcal/mol (**Cu-TS4a**), and *path b* is energetically unfavored with a rate determine barrier 33.8 kcal/mol (Fig. 7a, only key geometries was showed).

**[4+2] reaction path way**. Avcı et al. explored [4+2] mechanism with density functional theory (DFT), in their calculation, they simplified triisopropylsilyl (TIPS) group into trimethylsilyl (TMS) group and found that this reaction can proceeded through [4+2] stepwise mechanism[17]. We compared the rate determine barrier (RDB) of our proposed SMC mechanism with [4+2] stepwise mechanism under their model use different level of theory and found that the barrier of our proposed SMC mechanism lower than the [4+2] stepwise mechanism from 9.8 kcal/mol to 11.2 kcal/mol, which is a strong support that our proposed SMC mechanism is much more superior to [4+2] stepwise mechanism (see details in Supplementary Tables 1–3 and Supplementary Fig. 1). If we use the same reference point as their work (**L-Ag$^+$**, **1a** and **2a** as reference point) and no reduced model, the barrier of silver catalyzed [4+2] mechanism is 26.1 kcal/mol which is also in accordance with their work.

In order to further illustrate that our proposed SMC mechanism is more suited to silver(I) and copper(I) catalyzed IEDDA reaction, we evaluated the barrier of traditional [4+2] reaction pathway with relatively more stable π-complexes (**Ag-int1** and **Cu-int1**) as reference point.

With **Ag-int1** as zero point, the barrier of silver(I) catalyzed stepwise mechanism is 33.3 kcal/mol and concerted mechanism 34 kcal/mol (Fig. 8a).

The barrier of copper-catalyzed [4+2] cycloaddition is very high with stepwise mechanism 39.3 kcal/mol and the concerted mechanism 43.3 kcal/mol (Fig. 8b). Obviously, [4+2] cyclization pathway is not favored both in silver(I) and copper(I) catalyzed IEDDA reaction.

## Discussion

After formation of d$^{10}$ metal (Ag$^+$ or Cu$^+$) coordinated four-member ring complex **Ag-int3** or **Cu-int3**, we also evaluated the ring expansion and contraction way (Fig. 9). The rate determine barrier (RDB) of ring expansion is 27.5 kcal/mol and this process is energetically not favored compared with *path a* (RDB 20.3 kcal/mol). However, after formation of 1,2-NCOT, ring contraction to release N$_2$ is energetically unfavored with barrier as high as 38.2 kcal/mol which means 1,2-NCOT is dynamic stable[11,36–38]. The barrier of formation 1,2-NCOT is just between our proposed SMC mechanism (20.8 kcal/mol) and stepwise [4+2] mechanism (33.3 kcal/mol). If the silylium ion transfer step was inhibited, 1,2-NCOT may be generated. Therefore, our calculation implies a very concise way to synthesis 1,2-NCOT.

Rawal et al. also declared that the electron-rich ethoxy alkynes **2c** and ynamide **2d** (Fig. 1b) cannot react with phthalazine **1a** under the same condition as siloxy alkyne **2a**. According our proposed SMC mechanism, both alkyenes **2c** and **2d** have no silicane groups and formation of ketene species was inhibited. For ethoxy alkyne **2c** may possibility go through carbocation transfer to form ketene species after formation of four-member ring **M-int3** (**Ag-int3**, **Cu-int3**) similarities, we evaluated barrier of this process as high as 34.1 kcal/mol. For ynamide **2d**, it's evident that the protection group cannot form any ion. If we change protection group of ynamide **2d** into silane, this SMC pathway maybe also available to nitrogen-substituted alkynes. Actually electron-rich ethoxy alkynes **2c** and ynamide **2d** should be more facile to react with phthalazines **1a** if this d$^{10}$ metal-catalyzed IEDDA proceed through [4+2]

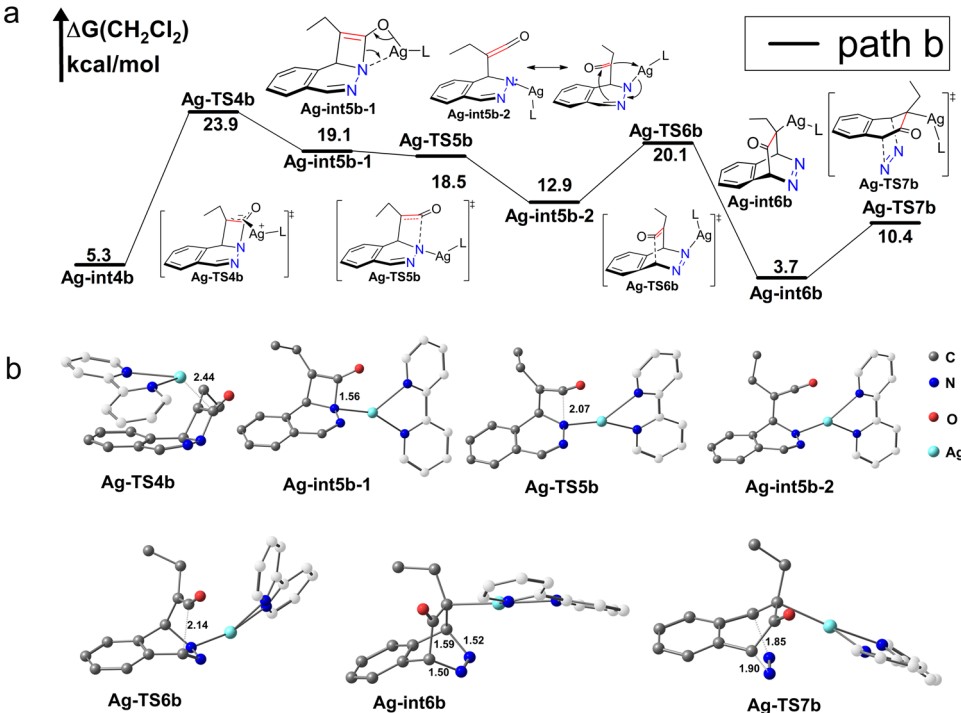

**Fig. 6 Ketone-enol tautomerism path way (*path b*).** L refers to 2,2′-bipyridine(bpy). Gibbs free energies are in kcal/mol. The hydrogen atom was omitted and carbon atom of ligand [2,2′-bipyridine(bpy)] was white cloaked for clarity. **a** Reaction profile; **b** Optimized geometries.

cycloaddition pathway. It's a strong support that protection group TIPS also take effect in this reaction.

For silver(I) and copper(I) catalyzed IEDDA, the difference of the rate determined barrier is 9.9 kcal/mol for path b, 6.5 kcal/mol for path a, and 6.0 kcal/mol for [4+2] path. In each case the barrier for copper(I) is higher which is in accordance with experiment. We think this is partly due to relative smaller radius and higher positive charge density of copper(I) atom, which lead to a stronger coordination of the substrate. According to our calculation, formation of **L-Cu$^+$** is 10 kcal/mol more exothermic than formation **L-Ag$^+$**(−33.0 kcal/mol vs −23.1 kcal/mol). Formation of the π-complex **Cu-int1** is much more exothermic than **Ag-int1**(−30.3 kcal/mol vs −60.8 kcal/mol, Fig. 3a). π-complex **Cu-int1** is more stable than **Ag-int1** in solution, which lead to a relatively lower reference point of copper(I) catalyzed IEDDA. This is main cause of relatively higher barrier of copper(I) catalyzed IEDDA. In addition, copper(I) atom is more prone to be oxidized, NBO charge distribution shows that there was evident electron transfer from copper(I) to siloxy alkyne which may also lead to stronger coordinate of copper(I) to siloxy alkynes (NBO charge distribution shows that the electron changed from −0.164 to −0.271 in the C2 and 0.293 to 0.248 in the C3 when siloxy alkyne **2a** coordinated with copper(I) to form **Cu-int1**, electron charge changed into −0.218 in C2 and 0.312 in C3 when siloxy alkyne **2a** coordinated with silver(I) to form **Ag-int1**).

Lewis acid-catalyzed IEDDA meet limited success which often confine to modification of the two π-electron components (dienophile or diene) for a better LUMO-HOMO matching. Our calculation provides a remarkably different perspective to access IEDDA cycloaddition product which is difficult obtain through traditional [4+2] pathway. In this reaction, high affinity of d$^{10}$ metal (Ag$^+$ or Cu$^+$) to alkynes and alkenes activate the [2+2] cycloaddition and transfer of TIPS$^+$ respectively. The electron extraction competition between d$^{10}$ metal and TIPS$^+$ lead to final product. In most silylium ion-catalyzed reaction, high acidity silylium ion should be prepared before experiment or generated

in situ which often need highly active reagent, this π-acid and hard acid in situ exchange scheme may enlighten in silylium ion chemistry. This mechanism also implies a concise way to synthesis 1,2-dinitrogen substituted cyclooctatetraene (1,2-NCOTs) which is quite difficult to obtain with current methods. Nitrogen-substituted alkynes such as ynamines and ynamides are inert to IEDDA reaction, our SMC pathway may overcome this difficulty by change protection group into silicane.

## Methods

**Model reaction.** Phthalazine **1a** and ethyl substituted siloxy alkyne **2a** was choose as the model reactions. 2,2′-bipyridine (bpy) was choose as the ligand in both silver(I) and copper(I) catalyzed IEDDA reactions. AgOTf and CuOTf was choose as catalyst salt respectively (Fig. 3a). Dichloromethane (DCM) was choose as the solvent.

**Computational details.** Density function calculations (DFT) were performed with the Gaussian 16 program package[39]. All structural optimizations were performed with M062x density functional[40] with effective core potential (ECP) def2-TZVP[41] for silver and 6-31 G(d,p) basis set[42] for copper and other atoms (Optimized geometries see Supplementary Data 1–63). Single point energy was calculated at B2PLYPd3(BJ)/def2tzvp level of theory[43]. Vibrational frequency calculations were carried out at same level of theory as geometries optimization to verify that the optimized geometries is an energy minimum or a transition state and to provide thermal corrections for Gibbs free energies and enthalpies at 298.15 k in 1 atm. The intrinsic reaction coordinate (IRC) calculation was conducted to connect the transition structures with the corresponding reactants and products[44] (Supplementary Figs. 5–11). Solvation calculations were carried out with CPCM solvation model[45] at same level of theory with dichloromethane (DCM, ε = 8.93) as solvent. All the geometries were optimized in the gas phase except transition state of desilication step and inverse process which is optimized in solvent for electrostatic effect and complex conformation of triisopropylsilyl (TIPS) group make it very hard to load these structures in the gas phase.

**Functionals test.** It is commonly accepted that M06 or M06-d3 describes better the features of transition metals. We also optimized the key geometries of SMC pathway with both M06 and M06-d3 functionals. The M06 and M06-d3 optimized geometries are very similar (Supplementary method A). We corrected single point energy of M06-d3 optimized geometries under B2PLYPd3(BJ)/def2tzvp level of theory. Although M062x and M06-d3 predict slightly different geometries, the final reaction profiles based on M062x and M06-d3 optimized geometries are quite similar (Supplementary Figs. 2 and 3).

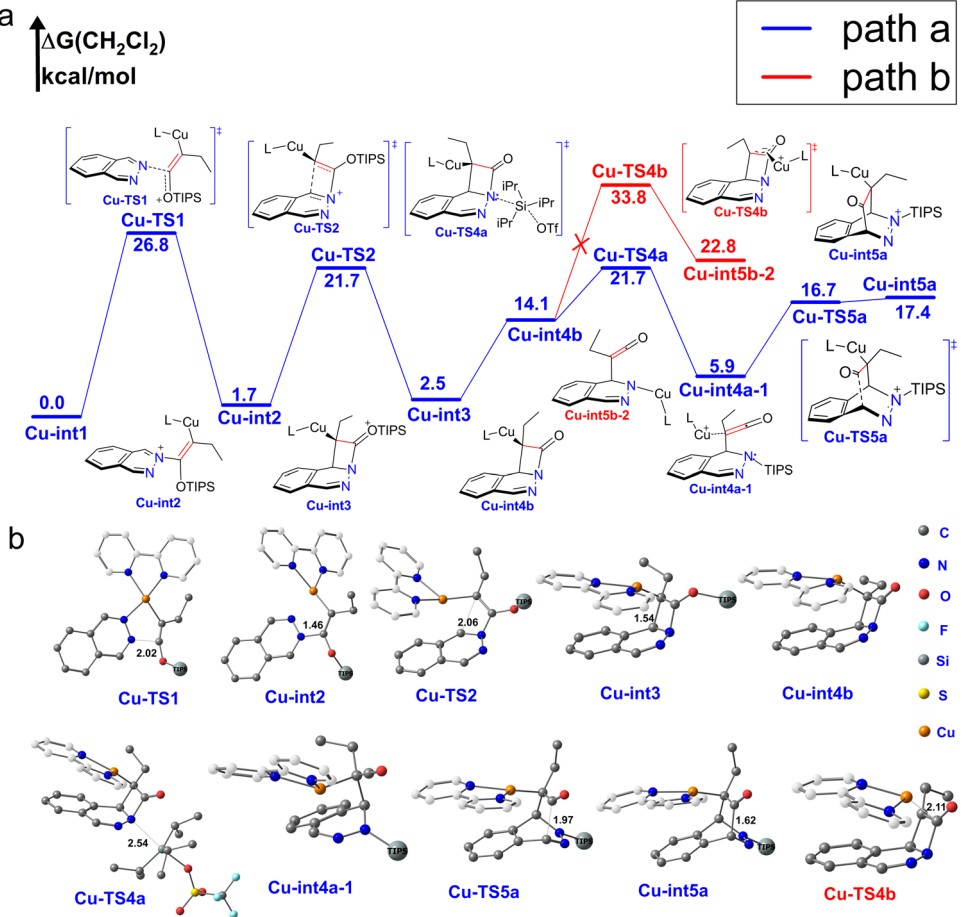

**Fig. 7 Copper(I) catalyzed IEDDA.** L refers to 2,2'-bipyridine(bpy). Gibbs free energies are in kcal/mol. The hydrogen atom was omitted and carbon atom of ligand [2,2'-bipyridine(bpy)] was white cloaked for clarity. Triisopropylsilyl (TIPS) is simplified into big ball except desilication process and versa. **a** Reaction profile; **b** Optimized geometries.

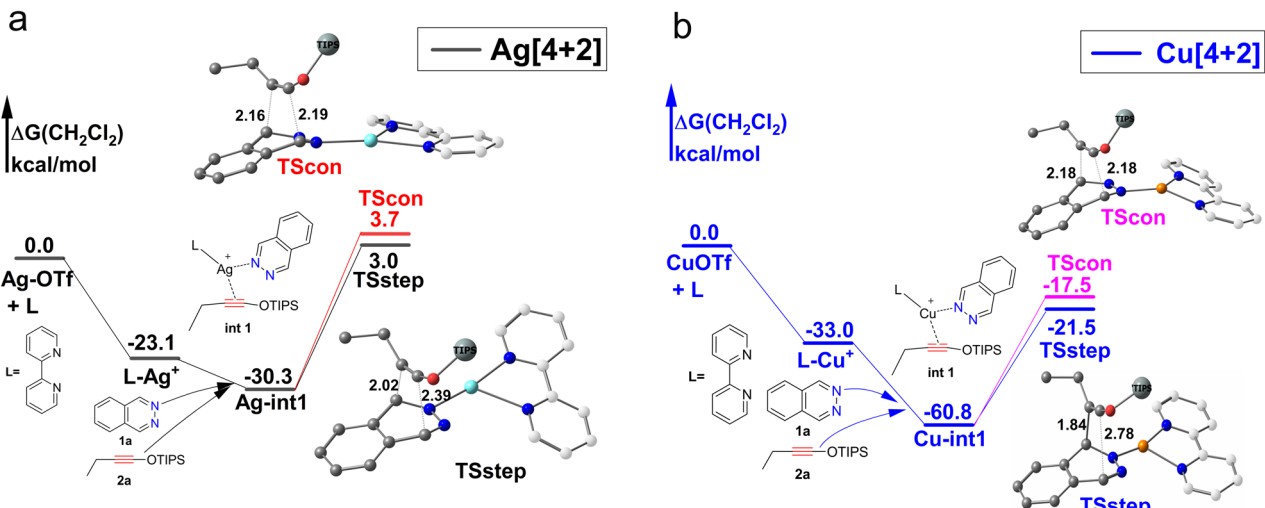

**Fig. 8 [4+2] cycloaddition pathway.** The hydrogen atom was omitted, triisopropylsilyl (TIPS) group was simplified to big ball and the carbon atom of ligand [2,2'-bipyridine(bpy)] was white cloaked for clarity. Gibbs free energies are in kcal/mol. **a** Silver (I) catalyzed 4+2 pathway; **b** Copper (I) catalyzed [4+2] pathway.

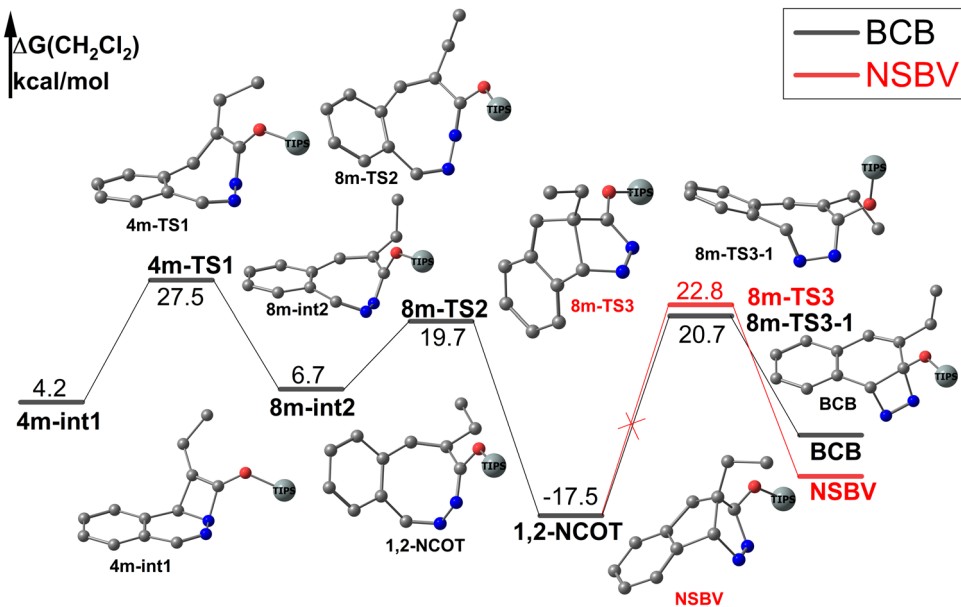

**Fig. 9 Reaction profile of formation 1,2-dinitrogen substituted cyclooctatetraene(1,2-NCOT) if silylium ion transfer step is inhibited.** Gibbs free energies are in kcal/mol. The hydrogen atom was omitted and triisopropylsilyl (TIPS) group was simplified to big ball for clarity.

## Data availability

The authors declare that all the other data supporting the findings of this study are available within this paper, its Supplementary Information file and Supplementary Data 1–63.

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

## Acknowledgements
Financial support from National Key Research and Development Program of China (No. 2016YFB0600301), National Natural Science Foundation of China (21673224, 21873096), Chinese Academy of Sciences (XDB17010200), and Dalian National Laboratory for Clean Energy (DNL180204) are acknowledged.

## Author contributions
H.-D.W. conceived, designed the study and performed the calculations. H.-J.F. supervised the project and contributed the design of the research and oversaw the writing of the manuscript.

## Competing interests
The authors declare no competing interests.
