## [Peer Review File · Communications Chemistry]

Reviewers' comments:

Reviewer #1 (Remarks to the Author):

See attached

Reviewer #2 (Remarks to the Author):

In this manuscript, Wang and Fan report a mechanistic study by DFT means on the silylium ion mediated inverse-electron mediated Diels-Alder reaction.

The theoretical study is well done technically, and the conclusions are adequate. In my opinion, the work is sound, especially because it corrects previously proposed mechanisms.

However, I believe that this work is not publishable in its present form, for the following reasons:

1) Both grammar and spelling contain many errors, making difficult to follow the discussion at some points of the paper. It is crucial to make a thorough correction of the paper.

2) The authors have chosen M06-2X for the structural optimization, although it is known that M06 functional describes better the features of transition metal complexes, as is the case for Cu and Ag catalysts. If they believe M062X is adequate, they should explain why, or otherwise reconsider to use M06, at least for the crucial schemes.

3) According to the experimental work by Rawal, Cu seems to be more efficient in promoting the reaction than Ag. However, the calculations point to the opposite situation, when comparing the activation barrier in Figures 2 and 6. Explain it or elaborate it further

4) There are many mentions during the discussion to the formation of the 1,2-NCOT product, which is regarded as a dynamically stable compound. According to the authors it could appear in certain conditions. But it is not clear to me if this compound has ever been isolated in this type of reactions, or whether it is just a theoretical prediction without any experimental support. Please clarify.

Thus, the work is too preliminary for publication in this form

Reviewer #3 (Remarks to the Author):

In this manuscript, Fan and co-workers reported their investigations on the mechanism of silver(I)- and copper(I)-catalyzed [2+2] cycloaddition reactions between 1,2-diazines and siloxyalkynes by density function theory (DFT). The results showed the migration of silylium ion [triisopropylsilyl ion (TIPS+)] controled the reconstruction of four-member ring to give the final product. This work not only disclosed that the SMC mechanism is more favorable for this silver(I) and copper(I) catalyzed IEDDA reaction, but also presented a powerful method for the synthesis of 1,2-dinitrogen substituted cyclooctatetraene (1,2-NCOTs). In addition, the paper was well organized and written. Overall, this work is innovative and interesting, and this reviewer would like to recommend the acceptance for Communications Chemistry after addressing the following concerns:

1. The reaction profiles of ketone-enol tautomerism pathway(Figure 4a) showed that the direct intramolecular isomerization via Ag-TS4b has an activation free energy of 18.6 kcal/mol (path b); however, in the copper(I) catalyzed IEDDA reaction (Figure 5a), path b is energetically unfavored due to the overall activation free energy of 33.8 kcal/mol (from Cu-int1 to Cu-TS4b). What causes this difference in energy barriers should be explained.

2. From four-member ring intermediate Ag-int4a to six-member ring intermediate Ag-int5a, there actually experience twice electron extraction competition between Ag-C and Si-N. The reason for the charge transfer should be explained in detail.

3. Generally, release nitrogen go through concerted reaction pathway, in this reaction, release N2 go through stepwise mechanism, the author thinks this can be attributed to high positive electron density of silylium ion (TIPS+), the role of silylium ion (TIPS+) should be explained in detail.

4. The bond angle of $\angle C2C3O$ is 178.04°in Ag-int1. The geometric data of $\angle C2C3O$ listed in the right is wrong (Figure 1).

5. In the computational details, the basis set for Ag is described. However, the basis set for Cu is not specified and needs to be supplemented.
6. Direct dissociation of L-Ag+ from Ag-int4b need to be displayed in supporting information.
7. IRC calculation results / graphics for key transition states should be given in supporting information.
8. One spelling mistake: "Ag-int1" in the paragraph below figure 1.

In this manuscript, Wang and Fan describe their computational studies on the mechanism of the formal inverse electron-demand Diels-Alder (IEDDA) reactions between phthalazines and siloxy alkynes catalyzed by Ag(I) and Cu(I) complexes. The experimental results for these transformations were reported by Rawal, Kozmin and co-workers in 2012 and 2014 (references 17 and 18 in the current manuscript). A related computational study on the same silver-catalyzed transformation was reported this year by Catak, Aviyente, Dedeoglu and co-workers (*ChemCatChem*, **2020**, *12*, 366; ref. 24). In this latter work, the researchers concluded that the calculations supported the stepwise pathway for the [4+2] cycloaddition over the concerted mechanism. However, in the current manuscript, the authors propose a substantially different mechanism, which involves an initial [2+2] cycloaddition, followed by a silylium ion transfer. This mechanism is definitely intriguing. While such IEDDA reactions of azines are highly valuable as they lead to the synthesis of a variety of (hetero)cycles, catalytic variants are highly limited. In this context, understanding the mechanisms of these catalytic IEDDA reactions is crucial for the development of new and better catalytic reactions. Therefore, an in-depth computational analysis of such catalytic IEDDA reactions will be of interest to organic chemists working in the areas of catalysis and heterocyclic chemistry. However, as listed below, several aspects of the current study arise questions and require more detailed analysis.

- The reactions of certain substituted 1,2-diazines with ynamines were shown to proceed via [2+2] cycloaddition, the ring opening of which gave diaza-cyclooctatetraene derivatives. The authors are recommended to cite the following articles on this topic: 1) *Chem. Pharm. Bull.* **1991**, *39*, 1713-1718. 2) *Heterocycles*, **1996**, *43*, 199-204.
- There are some major differences between the results obtained in this work and in the *ChemCatChem* article mentioned above (ref 24): For instance, in Table 1, the uncatalyzed reaction was calculated to have a reaction barrier of 40.0 kcal/mol, whereas the barriers for the Ag(I)-catalyzed concerted and stepwise mechanisms were calculated to be 34.0 and 33.3 kcal/mol, respectively. However, in ref. 24, the Ag(I)-catalyzed concerted and stepwise pathways were computed to have barriers of

32.2 and 25.0 kcal/mol, respectively. In addition, the results for the complexation of Ag(I) to the ligand (2,2'-bipyridine), phthalazine and siloxy alkyne are highly different in the two studies. It is recommended that the authors provide an explanation for these differences.

- Scheme 2a and Figure 2a: The first step of the stepwise [2+2] cycloaddition is shown as the attack of phthalazine nitrogen to C1 of siloxy alkyne activated by the Ag(I) complex (conversion of **Ag-int1** to **Ag-int2**). When the electron-rich nature of siloxy alkynes and electron-deficient nature of phthalazine are considered, an alternative initial step can be proposed to occur via the attack of the siloxy alkyne from C2 to the C1 of phthalazine. However, the authors did not discuss this possibility. It will be better if the authors compare the Gibbs energies for these two possible pathways.
- Scheme 2a and Figure 2a: The conversion of **Ag-int2** to **Ag-int3** seems problematic from an organic chemistry perspective. In this transformation, the silyl enol ether moiety was proposed to attack the iminium carbon intramolecularly, rather than the C-Ag bond. Given the highly electron-rich nature of such carbon-metal bonds, the C-Ag bond seems a better nucleophile candidate to attack the iminium carbon. Moreover, the proposed product of this step, **Ag-int3**, continues to have a C-Ag bond. Such a C-Ag bond is present in most the intermediates, which look counter-intuitive. Similar arguments can be made for the Cu(I)-catalyzed reaction.
- Figure 4a: In **Ag-int5b-1**, the carbonyl carbon makes five bonds. In addition, the transformation of **Ag-int5b-1** to **Ag-int5b-2** is difficult to understand.
- Page 4, "Model reaction" section: "1,10-phenanthroline" should be "phthalazine".
- Some references (such as references 5, 19, 20 and 24) do not have page numbers and/or volume numbers.
- There are many grammatical errors and typos throughout the manuscript, which make the text difficult to follow and understand.

Comments of reviewer 1:

In this manuscript, Wang and Fan describe their computational studies on the mechanism of the formal inverse electron-demand Diels-Alder (IEDDA) reactions between phthalazines and siloxy alkynes catalyzed by Ag(I) and Cu(I) complexes. The experimental results for these transformations were reported by Rawal, Kozmin and coworkers in 2012 and 2014 (references 17 and 18 in the current manuscript). A related computational study on the same silver-catalyzed transformation was reported this year by Catak, Aviyente, Dedeoglu and co-workers (*ChemCatChem*, **2020**, *12*, 366; ref. 24). In this latter work, the researchers concluded that the calculations supported the stepwise pathway for the [4+2] cycloaddition over the concerted mechanism. However, in the current manuscript, the authors propose a substantially different mechanism, which involves an initial [2+2] cycloaddition, followed by a silylium ion transfer. This mechanism is definitely intriguing. While such IEDDA reactions of azines are highly valuable as they lead to the synthesis of a variety of (hetero)cycles, catalytic variants are highly limited. In this context, understanding the mechanisms of these catalytic IEDDA reactions is crucial for the development of new and better catalytic reactions. Therefore, an in-depth computational analysis of such catalytic IEDDA reactions will be of interest to organic chemists working in the areas of catalysis and heterocyclic chemistry. However, as listed below, several aspects of the current study arise questions and require more detailed analysis.

Reply: We are very grateful to reviewer 1's comment and valuable suggestions.

1. Question: The reactions of certain substituted 1,2-diazines with ynamines were shown to proceed via [2+2] cycloaddition, the ring opening of which gave diazacyclooctetraene derivatives. The authors are recommended to cite the following articles on this topic: 1) *Chem. Pharm. Bull.* **1991**, *39*, 1713-1718. 2) *Heterocycles*, **1996**, *43*, 199-204.

Reply: The recommended reference is very valuable to our work. We have referred it in the introduction section (refer 12,13) with a brief discussion.

2. Question: There are some major differences between the results obtained in this work and in the *ChemCatChem* article mentioned above (ref 24): For instance, in Table 1, the uncatalyzed reaction was calculated to have a reaction barrier of 40.0 kcal/mol, whereas the barriers for the Ag(I)-catalyzed concerted and stepwise mechanisms were calculated to be 34.0 and 33.3 kcal/mol, respectively. However, in ref. 24, the Ag(I)-catalyzed concerted and stepwise pathways were computed to have barriers of 32.2 and 25.0 kcal/mol, respectively. In addition, the results for the complexation of Ag(I) to the ligand (2,2'-bipyridine), phthalazine and siloxy alkyne are highly different in the two studies. It is recommended that the authors provide an explanation for these differences.

Reply: There were two differences between our work and the *ChemCatChem* article (*ChemCatChem*, **12**, 366–372 (2020), ref 26 in revised manual script): firstly, In the ref 26, the author reduced more crowded protection group triisopropylsilyl (TIPS) into trimethylsilyl (TMS). Secondly, in ref 26 the authors choose **L-Ag+**, **sub 1** and **sub 2** as reference point, other than a more thermal stable complex **Ag-int1** before the transition state (see SI part I). If we use the same reference point as ref 26, the barrier of stepwise [4+2] mechanism is 26.1kcal/mol which is very

similar with their work (see figure S1 in the SI and result section part III of the revised manuscript). Also, if they use the same reference as us, the barrier for the stepwise mechanism would be 31.3 kcal/mol which is quite close to our value.

The energy difference of concerted and stepwise mechanism is 7.2kcal/mol in ref 26, and only 0.7kcal/mol in our work. To make clear the cause of this difference, we have compared reaction profiles with the same model as ref26 (with TMS as protection group) under wb97xd/def2tzvp//M06-2x/def2tzvp,6-31G(d,p) level of theory with the same reference point as ref 26. The reaction barrier obtained from wb97xd/def2tzvp//M06-2x/def2tzvp,6-31G(d,p) level of theory is in accordance with ref26 (the uncatalyzed [4+2] barrier 46.6kcal/mol vs 45.2 in ref26, stepwise [4+2] mechanism 27.0 vs 25.0 in ref26, concerted mechanism 32.9kcal/mol vs 32.2kcal/mol in ref26). We further calculated reaction barrier with the same model under B2PLYPd3(BJ)/def2tzvp//M06-2x/def2tzvp,6-31G(d,p) level of theory and the barrier is in accordance with our work (uncatalyzed [4+2] mechanism 41.6kcal/mol vs 40.0kcal/mol in our work, stepwise [4+2] mechanism 25.4kcal/mol vs 26.1kcal/mol in our work, concerted [4+2] mechanism 28.1kcal/mol vs 26.8kcal/mol in our work). It turned out that energy differences of concerted and stepwise mechanism mainly come from theoretical method to get the electronic energy (see details in table S2). For both uncatalyzed and catalyzed mechanism, the concerted barriers from wb97xd are higher than that from B2PLYPd3(BJ) by about 5kcal/mol, while the stepwise barriers are close. Since uncatalyzed mechanism is a pure organic system where double hybridized functional B2PLYPd3 is expected to work very well, we think although Wb97xd is often quite suitable for metal catalyzed system, in this reaction the barrier of concerted mechanism may be overestimated. To avoid such functional induced errors, we further compared rate determine barrier of our proposed SMC mechanism and [4+2] mechanism with different functional (wb97xd, pbe0, b3lyp-d3, pbe0-d3, m06-d3, b2plyp, mpw2plyp, b2plyp-d3, b2plypd3(BJ), B3lyp see details in SI table S3) and found the rate determine barrier of SMC mechanism lower [4+2] mechanism 9.2-16.2kcal/mol (with **TIPS** group as used in this work. The value will be 9.8-11.2kcal/mol, as shown in table S1, if **TMS** group is used as in ref26) among all the tested functional. Our proposed SMC mechanism is more favored than [4+2] mechanism for every functional that has been tested.

We have discussed the difference between two studies detailed in SI (part I) and in the revised manuscript (result section part III and introduction section).

3. Question: Scheme 2a and Figure 2a: The first step of the stepwise [2+2] cycloaddition is shown as the attack of phthalazine nitrogen to C1 of siloxy alkyne activated by the Ag(I) complex (conversion of **Ag-int1** to **Ag-int2**). When the electron-rich nature of siloxy alkynes and electron-deficient nature of phthalazine are considered, an alternative initial step can be proposed to occur via the attack of the siloxy alkyne from C2 to the C1 of phthalazine. However, the authors did not discuss this possibility. It will be better if the authors compare the Gibbs energies for these two possible pathways.

Reply: Thanks for the comments. We have tested the C2 attack point and the barrier is 32.4kcal/mol, which is much higher than C1 (19.5kcal/mol) case. We think the lower barrier of C1 attack mainly due to the positive charge on C induced by Ag(I) can be stabilized by O atom through conjugated effect.

We have discussed this C2 attack pathway in the revised manuscript (part II in the results section).

4. Question: Scheme 2a and Figure 2a: The conversion of **Ag-int2** to **Ag-int3** seems problematic from an organic chemistry perspective. In this transformation, the silyl enol ether moiety was proposed to attack the iminium carbon intramolecularly, rather than the C-Ag bond. Given the highly electron-rich nature of such carbon-metal bonds, the C-Ag bond seems a better nucleophile candidate to attack the iminium carbon. Moreover, the proposed product of this step, **Ag-int3**, continues to have a C-Ag bond. Such a C-Ag bond is present in most the intermediates, which look counterintuitive. Similar arguments can be made for the Cu(I)-catalyzed reaction.

Reply: That's a very good suggestion. Indeed, from **Ag-int2** to **Ag-int3** the electron transfers from electron-rich C-Ag bond to electron-deficient C-N bond. In **Ag-int2** there is a strong C-Ag bond, while in the **Ag-int3** Ag and C is only weakly coordinated. Actually, there are two resonance Lewis structure of **Ag-int3** and only one of them was showed in previous manuscript. We have added another resonance Lewis structure in the figure 2a. We have made a through correction of such vague Lewis structure to avoid confusion (**Ag-TS4a** and **Ag-TS4a-1** in figure 3a. **Ag-int5b-1**, **Ag-int6b** and **Ag-TS7b** in figure 4a).

5. Question: Figure 4a: In **Ag-int5b-1**, the carbonyl carbon makes five bonds. In addition, the transformation of **Ag-int5b-1** to **Ag-int5b-2** is difficult to understand.

Reply: Thanks for your valuable comment. There was some error in the Lewis structure we have present. We have corrected the Lewis structure of **Ag-int5b-1** and labeled some arrows in the **Ag-int5b-1** for better understand of electron transfer.

6. Question: Page 4, "Model reaction" section: "1,10-phenanthroline" should be "phthalazine".

Reply: Thank you for your comment, we have corrected the error.

7. Question: Some references (such as references 5, 19, 20 and 24) do not have page numbers and/or volume numbers.

Reply: Thank you for comment, we have corrected all the reference.

8. Question: There are many grammatical errors and typos throughout the manuscript, which make the text difficult to follow and understand.

Reply: Thank you for comment. We have checked the grammatical and spelling error entirely in the revised manuscript.

Comments of reviewer 2:

In this manuscript, Wang and Fan report a mechanistic study by DFT means on the silylium ion mediated inverse-electron mediated Diels-Alder reaction.

The theoretical study is well done technically, and the conclusions are adequate. In my opinion, the work is sound, especially because it corrects previously proposed mechanisms.

However, I believe that this work is not publishable in its present form, for the following reasons:

Reply: We are greatly appreciated to the comments and constructive suggestions.

1. Question: Both grammar and spelling contain many errors, making difficult to follow the

discussion at some points of the paper. It is crucial to make a thorough correction of the paper.

Reply: Thank you for comment. We have made a thorough correction of our manuscript.

2. Question: The authors have chosen M06-2X for the structural optimization, although it is known that M06 functional describes better the features of transition metal complexes, as is the case for Cu and Ag catalysts. If they believe M062X is adequate, they should explain why, or otherwise reconsider to use M06, at least for the crucial schemes.

Reply: We have reoptimized geometries of Ag catalyzed IEDDA with M06 functional and M06-d3 functional. The difference between M06 and M06-d3 functional is relatively small (the average difference between **C-Ag** bond 0.004 Å, **C-N** bond within 0.001 Å, bond angle within 0.31° among the tested geometries, see details in figure S2).

We further compared the parameters between M06-d3 and M06-2X optimized geometries of Ag catalyzed IEDDA. The difference between the two functional is relatively acceptable (the average difference of **N-Ag** bond 0.04Å, average difference of **C-Ag** bond 0.1Å, average differences of bond angle 1.25°. see details in figure S2). To evaluate the influence of geometry difference to the reaction barrier, we further calculated Gibbs energy profile of our proposed SMC mechanism under B2PLYPd3(BJ)/def2tzvp//M06-d3/def2tzvp,6-31G(d,p) level of theory and the result is very similar with our original work (the difference of rate determine barrier is only 0.2kcal/mol, and the averaged difference for transition states and intermediates is 0.46kcal/mol, see details in figure S3).

For Cu catalyzed IEDDA, the difference between the two functional is relatively smaller than Ag case (the averaged difference between **C-Cu** bond 0.07 Å, **C-N** bond within 0.06 Å, bond angle within -0.65° among the tested geometries, see details in figure S2). To evaluate the impact posed by these geometries differences on reaction energy path way, we also calculated Gibbs energy profile of Coper(I) catalyzed SMC mechanism under B2PLYPd3(BJ)/def2tzvp//M06-d3/6-31G(d,p) level of theory and the result is also very similar with our previous work (the difference of rate determine barrier is 1.4kcal/mol, and the averaged difference is 0.25kcal/mol. see details in figure S4).

Therefore, although M062x and M06-d3 predict slightly different geometries, our final reaction energy profile based on M062x and M06-d3 geometries are quite similar. In the ref 26 (*ChemCatChem*. **12**, 366–372 (2020)), the author have tested a series of functional for geometries optimization [include GGA(PBE), Meta-GGA (M06 L), Hybrid-GGA (ω B97X-D, MPW1 K,PBE0,B3LYP, CAM-B3LYP-D3) and Hybrid-Meta-GGA (M06,MPWB1 K, BMK, BMK-D3)] and found that M062x is suitable for the current system. Recently, a detailed benchmark investigating on the transition metal reaction barrier heights in organometallic reactions including 3d–5d transition metals has also shown that M06-2X performs surprisingly well and even, M06-2X outperforms M06 (M. A. Iron, T. Janes, J. Phys. Chem. A. 2019, 123, 3761–3781). So, we consider the M06-2x functional is also appropriate for both Ag and Cu catalyzed IEDDA.

For the above reasons, we have chosen to keep the B2PLYPd3(BJ)/def2tzvp//M06-2x/def2tzvp,6-31G(d,p) profile in the manuscript, and listed the reaction profiles obtained under B2PLYPd3(BJ)/def2tzvp//M06-d3/def2tzvp,6-31G(d,p) level of theory in the SI.

3. Question: According to the experimental work by Rawal, Cu seems to be more efficient in promoting the reaction than Ag. However, the calculations point to the opposite situation, when comparing the activation barrier in Figures 2 and 6. Explain it or elaborate it further.

Reply: Thanks for the question. We have carefully evaluated Ag and Cu catalyzed IEDDA and found that Ag performs better than Cu in the same condition. In the Ag catalyzed IEDDA (J. Am. Chem. Soc. 2012, **134**, 9062–9065. Table 1, entry 7), the yield reach to 90% after 2h. In the Cu catalyzed IEDDA (Org. Lett. 2014, **16**, 3236–3239. Table 1, entry 6) the yield reaches only 34% after 24h in the very similar condition, our results is in accordance with Rawal's work. In the Cu catalyzed IEDDA, Rawal et.al finally choose Cu(MeCN)₄PF₆ as catalyst salt which reach 83% yield after 4h which is the best result of Cu catalyzed IEDDA (Org. Lett. 2014, **16**, 3236–3239. Table 1, entry 11). For the silver, the best result is 89% yield after 1.5h when use AgNTf as catalyst salt and the catalyst loading is much lower than Cu case (only one-tenth as Cu case, J. Am. Chem. Soc. 2012, **134**, 9062–9065. Table 1, entry 14). We have chosen AgOTf and CuOTf as catalyst salt for Ag and Cu catalyzed IEDDA for convenient, and this could also introduce some differences compared to the experiments. Rawal also reported that Cu can catalyze IEDDA between pyridopyridazines and siloxy alkynes (Org. Lett. 2014, **16**, 3236–3239. Figure1, entry 16) when refluxed in dichloroethane (DCE) while Ag was inefficient to this reaction. The temperature of this reaction as high as 83.5°C and we think this reaction may not proceed through our proposed SMC mechanism.

4. Question: There are many mentions during the discussion to the formation of the 1,2-NCOT product, which is regarded as a dynamically stable compound. According to the authors it could appear in certain conditions. But it is not clear to me if this compound has ever been isolated in this type of reactions, or whether it is just a theoretical prediction without any experimental support. Please clarify.

Thus, the work is too preliminary for publication in this form

Reply: At the beginning we did not notice clear experimental evidences for Nitrogen-substituted COT as unexpected byproduct of IEDDA, and it is just a theoretical prediction. Thanks for Reviewer 1, he recommended us two excellent example 1) *Chem. Pharm. Bull.* **1991**, *39*, 1713-1718. 2) *Heterocycles*, **1996**, *43*, 199-204. In the recommended reference, substituted 1,2-diazines with ynamines were shown to proceed both via [2+2] cycloaddition and [4+2] IEDDA cycloaddition at the same time, the [2+2] cycloaddition followed with ring expansion to give diazacyclooctatetraene derivatives. They obtained [4+2] IEDDA product and diazacyclooctatetraene derivatives with comparable yield.

We also found some example of direct [2+2] between alkynes and aromatic rings, which then followed by ring expansion to obtain COT similarities. The yield of such reaction is relatively low. 1) *Chem. Commun.*, 2007, 5119–5133. 2) *Tetrahedron* **57** (2001) 7575-7606. 3) *Can. J. Chem.* **81** 37–44 (2003). 4) *Journal of Organometallic Chemistry* **693** (2008) 894–898. So, this prediction do have some experimental supports, and it indicates a very concise way to construct Nitrogen-substituted COT similarities.

Comments of reviewer 3:

In this manuscript, Fan and co-workers reported their investigations on the mechanism of silver(I)-

and copper(I)-catalyzed [2+2] cycloaddition reactions between 1,2-diazines and siloxyalkynes by density function theory (DFT). The results showed the migration of silylium ion [triisopropylsilyl ion (TIPS⁺)] controlled the reconstruction of four-member ring to give the final product. This work not only disclosed that the SMC mechanism is more favorable for this silver(I) and copper(I) catalyzed IEDDA reaction, but also presented a powerful method for the synthesis of 1,2-dinitrogen substituted cyclooctatetraene (1,2-NCOTs). In addition, the paper was well organized and written. Overall, this work is innovative and interesting, and this reviewer would like to recommend the acceptance for Communications Chemistry after addressing the following concerns:

Reply: We are very grateful to comment and good suggestion.

1. Question: The reaction profiles of ketone-enol tautomerism pathway(Figure 4a) showed that the direct intramolecular isomerization via Ag-TS4b has an activation free energy of 18.6 kcal/mol (path b); however, in the copper(I) catalyzed IEDDA reaction (Figure 5a), path b is energetically unfavored due to the overall activation free energy of 33.8 kcal/mol (from Cu-int1 to Cu-TS4b). What causes this difference in energy barriers should be explained.

Reply: Thanks for the nice comment. There was some misleading due to our poor expression. We have rewritten the related paragraphs to avoid such misleading. The rate determine step of path b is ketone-enol tautomerism step, the barrier relative to zero point (**Ag-int1** or **Cu-int1**) of this step is 23.9kcal/mol (**Ag-TS4b**) for silver(I) and 33.8kcal/mol (**Cu-TS4b**) for copper(I) catalyzed IEDDA. Path b is energetically unfavored relative to Path a for both silver(I) and copper(I) catalyzed IEDDA.

For silver(I) and copper(I) catalyzed IEDDA, the difference of the rate determined barrier is 9.9 kcal/mol for path b, 6.5 kcal/mol for path a, and 6.0 kcal/mol for [4+2] path. In each case the barrier for copper(I) is higher which is in accordance with experiment. We think this is partly due to relative smaller radius and higher positive charge density of copper(I) atom, which lead to a stronger coordination of the substrate. According to our calculation, formation of **L-Cu⁺** is 10kcal/mol more exothermic than formation **L-Ag⁺** (-33.0kcal/mol vs -23.1kcal/mol). Formation of the π -complex **Cu-int1** is much more exothermic than **Ag-int1**(-30.3kcal/mol vs -60.8kcal/mol, figure 1). π -complex **Cu-int1** is more stable than **Ag-int1** in solution, which lead to a relatively lower reference point of copper(I) catalyzed IEDDA. This is main cause of relatively higher barrier of Cu catalyzed IEDDA. There was also some experiment confirmed that copper(I) π -complex is more stable than silver(I) π -complex (ref 29). In addition, copper(I) atom is more prone to be oxidized, NBO charge distribution shows that there was evident electron transfer from copper(I) to siloxy alkyne which may also lead to stronger coordinate of copper(I) to siloxy alkynes (NBO charge distribution shows that the electron changed from -0.164 to -0.271 in the C2 and 0.293 to 0.248 in the C3 when siloxy alkyne 2a coordinated with copper(I) to form **Cu-int1**, electron charge changed into -0.218 in C2 and 0.312 in C3 when siloxy alkyne 2a coordinated with silver(I) to form **Ag-int1**).

We have discussed this in detail in our revised manuscript in the discussion section and results section (part I).

2. Question: From four-member ring intermediate Ag-int4a to six-member ring intermediate Ag-int5a, there actually experience twice electron extraction competition between Ag-C and Si-N.

The reason for the charge transfer should be explained in detail.

Reply: Thanks for the comments. In the **Ag-int4a** the high positive charge of **Si-N** extracts the electron from electron-rich **Ag-C** bond, this finally led to ring open of **Ag-int4a** to form **Ag-int4a-1**. The driving force of this step is high positive charge of **Si-N** and high tension of four-membered ring of **Ag-int4a**. From silver(I) coordinated ketene species **Ag-int4a-1** to six-member ring **Ag-int5a**, the barrier is 10.7kcal/mol. Although high reactivity of ketene species and formation of **N=N** bond contribute to this step, this step is endothermic due to higher positive charge of **Si-N**, we think the driving force of this step is extreme high exothermic of release **N₂**. We have discussed this process in detail in result section (Part II, stage II)

3. Question: Generally, release nitrogen goes through concerted reaction pathway, in this reaction, release **N₂** go through stepwise mechanism, the author thinks this can be attributed to high positive electron density of silylium ion (**TIPS⁺**), the role of silylium ion (**TIPS⁺**) should be explained in detail.

Reply: We think the role of silylium ion in this step is to heterolysis **C-N** bond for its extremely high Lewis acidity.

In the **Ag-int5a** the length of **C-N** bond directly links to silylium ion (**TIPS⁺**) is 1.53Å, another **C-N** bond is 1.45Å, which is mainly caused by high positive charge of **TIPS⁺**. The relatively longer **C-N** bond induced by **TIPS⁺** in the **Ag-int5a** indicate its relatively easy heterolysis character. Furthermore, the electron rich **C-Ag** bond can donate its electron to the C atom after heterolysis, this is very important driving force of **C-N** bond heterolysis. Actually, this heterolysis process is also an electron extraction competition between **Ag-C** and **Si-N**, the positive charge transferred from **N-Si** to **C-Ag** in this process (NBO charge distribution shows that electron changed from 0.16 to 0.59 in **C-Ag** and 1.79 to 1.58 in **N-Si** from **Ag-int5a** to **Ag-int6a**). Because of these reasons, the barrier for heterolysis **C-N** bond and stepwise **N₂** release is only 2.9kcal/mol. If there is no **TIPS⁺** in this structure (see **Ag-int6b** in figure 4b), the reaction can go through concerted mechanism with a barrier (**Ag-TS7b**) 6.7kcal/mol.

We have discussed the role of Silylium ion detailed in the result section (Part II, stage II) and labeled some arrows in **Ag-int5a** in figure 3a for better understand.

4. Question: The bond angle of $\angle C2C3O$ is 178.04° in **Ag-int1**. The geometric data of $\angle C2C3O$ listed in the right is wrong (Figure 1).

Reply: thank you for correction, we have changed the data in the figure 1.

5. Question: In the computational details, the basis set for Ag is described. However, the basis set for Cu is not specified and needs to be supplemented.

Reply: We have added basis set of Cu in the computational details.

6. Question: Direct dissociation of **L-Ag⁺** from **Ag-int4b** need to be displayed in supporting information.

Reply: we have displayed this process in supporting information (part III, figure S5)

7. Question: IRC calculation results / graphics for key transition states should be given in supporting information.

Reply: We have listed IRC calculations of key transition states in the supporting information.

8. Question: One spelling mistake: “Ag-int1” in the paragraph below figure 1.

Reply: Thank you for correction, we have changed the spelling mistake.

REVIEWERS' COMMENTS:

Reviewer #1 (Remarks to the Author):

In their revised manuscript, the authors addressed all the points raised in a satisfactory manner. Therefore, I would recommend the acceptance of this revised manuscript for publication in 'Communications Chemistry'.

Reviewer #2 (Remarks to the Author):

The authors have satisfactorily responded to all my previous concerns and made the necessary changes. Thus, the manuscript deserves publication now as it is

Comments of reviewer 1:

In their revised manuscript, the authors addressed all the points raised in a satisfactory manner. Therefore, I would recommend the acceptance of this revised manuscript for publication in 'Communications Chemistry'.

Reply: We are very grateful to reviewer 1's very valuable suggestions and positive comment for our work.

Comments of reviewer 2:

The authors have satisfactorily responded to all my previous concerns and made the necessary changes. Thus, the manuscript deserves publication now as it is.

Reply: We are greatly appreciated to the positive comments and constructive suggestions of reviewer 2 to this work.